# Effects on Intestinal Mucosal Morphology, Productive Parameters and Microbiota Composition after Supplementation with Fermented Defatted *Alperujo* (FDA) in Laying Hens

**DOI:** 10.3390/antibiotics8040215

**Published:** 2019-11-09

**Authors:** Agustín Rebollada-Merino, Carmen Bárcena, María Ugarte-Ruiz, Néstor Porras, Francisco J. Mayoral-Alegre, Irene Tomé-Sánchez, Lucas Domínguez, Antonio Rodríguez-Bertos

**Affiliations:** 1VISAVET Health Surveillance Centre, Complutense University of Madrid, 28040 Madrid, Spain; agusrebo@ucm.es (A.R.-M.); cbarcena@ucm.es (C.B.); nestorpo@ucm.es (N.P.); fjmayoral@ucm.es (F.J.M.-A.); aitome@ucm.es (I.T.-S.); lucasdo@visavet.ucm.es (L.D.); arbertos@ucm.es (A.R.-B.); 2Department of Animal Health, Faculty of Veterinary Medicine, Complutense University of Madrid, 28040 Madrid, Spain; 3Department of Internal Medicine and Animal Surgery, Faculty of Veterinary Medicine, Complutense University of Madrid, 28040 Madrid, Spain

**Keywords:** fermented defatted *alperujo* (FDA), olive oil by-products, intestinal health, laying hens, histomorphology, microbiota

## Abstract

The olive oil sector is currently adapting its traditional function to also become a supplier of high-value by-products that possess antioxidant, anti-inflammatory and antimicrobial properties. In this study, we evaluated the effect of the fermented defatted *alperujo* (FDA) on the intestinal health of laying hens. The morphology of the duodenal and cecal mucosa, the composition of the intestinal microbiota and the productivity of a batch of laying hens were evaluated after FDA supplementation. At early life stages, significant differences (*p* < 0.001) were observed in duodenal villi height and in crypt depth of both the duodenum and the cecum in the FDA-supplemented group, indicating improved intestinal health in this group. Microbiota composition in the hatchery group supplemented with FDA had a higher abundance of Actinobacteria, Firmicutes and Proteobacteria, and higher bacterial diversity. During the production period, significant differences (*p* < 0.05) were observed in the number of broken eggs from the supplemented group. We conclude that FDA supplementation improves the absorption capacity of the intestinal mucosa and modifies the intestinal microbiota to favor a greater immune response, leading to an increase in egg production.

## 1. Introduction

The European Union, particularly the Mediterranean countries, is the main producer of olive oil worldwide [1]. Spain is by far the largest manufacturer: a total of 1,260,000 tons were produced in the 2017–2018 campaign [1,2], with a mean worth of about 1.9 billion euros per year [3]. The olive oil industry has a major economic impact in the producing countries, accounting for around 46 million jobs per production cycle (from seed to final product). This industry also has cultural and environmental implications [3,4].

The centrifugation process used in olive oil production generates by-product wastes, which is of great concern due to their polluting activity and, thus, significant environmental impact [4,5]. Nowadays, the olive oil production process consists of two centrifugation phases, which has resulted in a reduction of waste products. As a result of this process, a solid by-product called two-phase olive mill waste, sometimes referred to as olive cake or olive pulp but commonly known as *alperujo*, is obtained [6]. The production of approximately one ton of olive oil generates four tons of this by-product [4,6].

Despite its pollutant nature, olive-oil by-products have been described as having antioxidant, anti-inflammatory, antimicrobial and anti-tumoral proprieties [7]. Furthermore, a significant proportion of this by-product is comprised of non-soluble fibers, carbohydrates, high-quality fats and proteins [6]. Some of these bioactive compounds are polyphenols, primarily hydroxytyrosol and tyrosol [8]. These molecules act as antioxidants by inhibiting oxidative reactions thus protecting the cell from oxidative damage and as anti-inflammatories by mediating a reduction in cytokine secretion [8,9]. In addition, the phenols and polyphenols contained in olive oil by-products have antibacterial activity against Gram-positive bacteria like *Staphylococcus aureus* and Gram-negative bacteria like *Campylobacter* spp. [8,10,11,12].

Consequently, the olive oil industry is currently adapting to also become suppliers of high-value by-products, which may reduce the industry’s environmental impact [13]. One of the proposed uses of olive oil by-products is in animal feed supplementation, and several studies have tested their effects in various animal species including poultry [13]. In broiler chickens, feed supplemented with *alperujo* have been demonstrated to improve productive performance parameters [14,15], and enhance redox status in tissues [16,17]. In addition, *alperujo* has been shown to possess anti-coccidial proprieties [18]. In laying hens, dried olive pomace supplementation has been demonstrated to modulate inflammation and cholesterol content in eggs through affecting gene expression [19].

Supplementation with *alperujo*, however, has not been as extensively studied in laying hens. *Alperujo* supplementation does not appear to affect production performance in laying hens [20,21,22,23,24].

*Alperujo*, like other olive oil by-products, is a fat-rich compound. However, although possess beneficial effects, fat-rich compounds should be limited in the feed formulation to avoid their counterproductive effects in high percentages [14]. For this reason, in this study *alperujo* had first undergone a fermentation process to stabilize the raw material, a hydrolysis to decrease the total fat content, and desiccation and grinding process in order to adapt it to animal feed, obtaining fermented defatted *alperujo* (FDA).

To our knowledge, the impact of *alperujo* or their derived products on intestinal health has not been previously evaluated in laying hens. Intestinal health depends mainly on gut environment and diet [25]. This environment is, in turn, defined by the intestinal mucosa and microbiota, which constitute the main components of the gut barrier. The intestinal mucosa is the most extensive surface in the organism: it absorbs nutrients and is equipped with numerous mechanisms that constitute a first line of defense against potential hazards [26]. Likewise, an adequate microbiota composition seems to limit pathogenic bacterial colonization [27].

Improvement of the intestinal health in laying hens may optimize nutrient absorption and, thus, production performance. It may also induce changes in microbiota that could protect against pathogenic colonization and disease, decreasing the use of antimicrobial agents. Our study assesses changes in the intestinal mucosa and microbiota, as well as productivity, of FDA-supplemented laying hens compared with controls during their productive life in a commercial farm.

## 2. Results

### 2.1. Histological Study

The results of the histomorphometric analysis of the duodenal and cecal mucosa are shown in Table 1 and Figure 1. In hatchery-staged hens, duodenal villi height and crypt depth and cecal crypt number and depth were significantly higher in the FDA-supplemented group (*p* < 0.05). In hens at the production phase I stage, the only significant difference observed was an increase in duodenal villi number in the FDA-supplemented group. In production phase II hens, only cecal crypt depth was significantly higher in the supplemented hens.

### 2.2. Production Performance

The effects of FDA on production performance of laying hens are shown in Table 2. Hen production percentages, total eggs (in terms of number and weight) and feed consumption improved with FDA supplementation; however, the differences observed between the two groups was not significant (*p* > 0.05). Mortality rate was higher in the treatment group; however, the differences observed between the two groups were not significant (*p* > 0.05). The economic impact of the difference in production performance between the two groups was a 1.7% increase in egg sale profits and a 1.5% decrease in feed costs for the treated group. The percentage of cracked or broken eggs eliminated from production was significantly lower in the treated group (*p* < 0.05) (Figure 2).

### 2.3. Metagenomics

Bacteroidetes, Firmicutes, Actinobacteria and Proteobacteria were the main bacterial phyla identified in all samples (Figure 3). The other phyla detected at lower levels were grouped together and classified as “Rest”. The relative abundance (RA) of Bacteroidetes (43.72 versus 43.42% in treated versus control, respectively), Actinobacteria (10.72 versus 6.82%) and Proteobacteria (4.30 versus 2.58%) were higher in the treated group, while the relative abundance of Firmicutes was higher in the control group (38.67 versus 43.67%). Hens at the hatchery stage showed a higher relative abundance of Actinobacteria (28.39 versus 14.31%), Firmicutes (45.92 versus 39%) and Proteobacteria (8.29 versus 1.64%) in the treated group compared to the control one, whereas the relative abundance of Bacteroidetes was higher in the latter (17.37 versus 44.60%). In phase-I hens, only Actinobacteria (5.72 versus 4.58%) and Bacteroidetes (50.19 versus 16.86%) were more abundant in the treated group. In phase-II hens, Bacteroidetes (55.29 versus 49.47%) was still more abundant in the treated compared to control group, as was Proteobacteria (2.25 versus 1.98%).

## 3. Discussion

Olive oil industry wastes have emerged as highly valuable products due to their potential beneficial properties such as being antimicrobial and anti-inflammatory [7]. One modified by-product, the fermented defatted *alperujo* (FDA), contains molecules like polyphenols, which make it interesting as a potential supplement in animal feed [8], as demonstrated by the capacity of polyphenols-containing olive oil by-products to modulate gene expression in vivo [19]. FDA supplementation has not been previously used for animal feed, although *alperujo* supplementation has been tested in several species with the objective of improving performance or exploiting the product’s antioxidant capacity [28]. Iannaccone et al. [19] demonstrated that olive oil pomace supplementation in laying hens affects gene expression and thus enhance oxidative status and improves inflammatory response, which suggests that *alperujo* may contribute to hens welfare and health. However, the effects of this natural compound on intestinal health have not been studied extensively, at least in laying hens.

Nutrient absorption and innate immune response depend on the mucosal structure of the intestine. Therefore, we performed a histomorphometric analysis to evaluate the effects of FDA on the morphology of the intestinal mucosa. Although no previous studies have focused on the impact of olive oil by-products on the intestinal morphology of laying hens, several have assessed the effects of other plant-derived compounds, whose fiber and bioactive compounds content has shown to improve intestinal morphology in broilers, particularly after oligosaccharides supplementation [29]. Therefore, the high fiber, high-quality fat and phenolic content of FDA may enhance intestinal health in supplemented hens.

The majority of these studies focused on the small intestine, mainly the jejunum mucosa, with only a few evaluating the duodenum [30,31,32,33,34,35,36,37], despite its very high absorption potential [38]. In one case, supplementation with bamboo vinegar or polyunsaturated fatty acid from an extruded flax product resulted in a significant increase in duodenal villi height in aged hens, suggesting that diet composition affects the duodenal mucosal structure [29,36]. In another study, an increase in duodenal crypt depth was observed in 33-week-old hens supplemented with rapeseed expeller cake [34]. In these studies, changes induced by different substances on the intestinal mucosa were evaluated during the productive phase. In contrast, in our study, the impact of FDA was assessed during the production cycle of laying hens, from hatchery to late phases. The significant increase in duodenal villi height that we observed in supplemented pullets supposes a concomitant increase in absorptive area, as has been proposed by other authors [36], which, in turn, leads to increased productivity. Changes in intestinal morphology during the initial phases, which are thought to influence digestion and performance, are critical for later intestinal functions during the productive phase [39].

The large intestine, particularly the cecum, has been rarely assessed in intestinal morphometric studies. In a recent study on the effects of essential oils and organic acids, no significant differences were reported in these segments [31]. In our study, however, cecal crypt depth proved to be a relevant parameter in our assessment of the impact of FDA on mucosal morphology. The increase in cecal crypt depth in pullets and aged hens appears to not only improve nutrient absorption but also influence a non-specific immune response by better responding to potential superficial epithelial damage. Due to the function of intestinal crypts in epithelial renewal [38,40], an increase in crypt depth could favor fermentation, digestion and water absorption capacity, prevent the emergence of disease in pullets and extend productive life in aged hens.

The goals of livestock feed supplementation are species-dependent. However, for all species, production performance must be maintained at an acceptable level [12], and egg production is not an exception. Our analysis of production performance indicates that FDA-supplemented hens performed better than control-fed hens. Feed is one of the most important costs in animal production [25]; therefore, the increase in profit from the sale of eggs from the supplemented group may be, in part, attributed to a decrease in feed consumption. To our knowledge, shell hardness has not been previously correlated with any olive oil by-product feed supplement. The intestine has a central role on calcium absorption, and thus, influences calcium metabolism [41]. The increase of villi height and crypt depth in the intestine increases duodenal and cecal absorption capacity, and therefore, the positive impact on eggshell hardness observed here may be in relation to the augmented mineral and absorption in the intestine as previously suggested [30,42]. In addition, other authors have shown that oligosaccharides present in dietary fiber may increase mineral uptake, and thus, the high fiber content of FDA may be in relation with an improvement of calcium absorption in the intestine [29]. The smaller percentage of broken eggs observed in the supplemented group may have also contributed to the economic increase. However, further studies are needed to determine whether FDA directly or indirectly improves shell hardness.

It is important to study the molecular and metabolic mechanisms behind the productive parameters in order to correctly manage animal feed [19]. In poultry, as in other species, the intestinal microbiota collectively acts as a metabolic organ, facilitating nutrient absorption and the immune response against pathogens [31,43]. Microbiota is determined by host genes and the environment, with diet being one of the most important factors. Modifications in dietary composition may, therefore, induce changes in microbiota [31]. Microbiota composition in poultry, especially in laying hens, has become an important area of research in veterinary medicine [27]. Although some studies have focused on microbiota variation in laying hens at different phases [44,45,46,47] and under diverse production systems [47,48], analyses of microbiota in laying hens fed with plant-derived supplements is limited [49,50]. Given the main role of microbiota in fermentation and digestive processes in laying hens [47], we focused on its composition in the cecum in our analyses.

According to the literature, cecal microbiota variations in laying hens are expected during the production cycle [44,47]. Consistent with this, we observed phyla fluctuations in our analyses. The metagenomics results showed the predominance of Bacteroidetes and Firmicutes, as has been previously reported in other studies [27,40,44,46,47]. In addition, our results suggest a decrease in bacterial diversity over time in both groups. For instance, we observed a progressive and proportional increase in Bacteroidetes compared to the other phyla in treated adult laying hens. Similar results have been reported by other studies [44,46]. Decreased microbial diversity has been related to a reduction in the production of short chain fatty acids, which leads to reduced intestinal barrier function [48]. According to some studies and in contrast to adult hens, Firmicutes abundance in pullets is expected to be higher than that of Bacteroidetes [44,46]. Our results are in agreement with these previous studies. Moreover, we observed that the relative abundance of Firmicutes was higher than that of Bacteroidetes in treated animals at the early hatchery phase, suggesting greater bacterial diversity in animals supplemented with FDA. Finally, we observed that Actinobacteria comprised a significant proportion of the microbiota at the hatchery stage but progressively decrease over time. Our results contrast with other studies in which Actinobacteria was considered a minority phylum, accounting for less than 1% of the total bacterial community [44,49]. In our study, the proportion of Actinobacteria was higher in treated animals, up to 25 weeks-old, compared with the control group. Similarly, the abundance of Proteobacteria, which progressively decreased over time, was present at higher levels in the treated group during the first post-hatching weeks. Although the relative abundance of Proteobacteria was higher in pullets, it was far from the 50% reported in the cecum of one-week-old laying hens [44].

## 4. Materials and Methods

### 4.1. Ethical Approval

This project has been carried out in accordance with animal welfare standards for the species with ethical approval from the Complutense University of Madrid and the Community of Madrid (PROEX 152/19).

### 4.2. Animals and Rearing Conditions

The study was performed in a commercial farm using a whole batch of Hy-Line 2015 laying hens (122,250 animals in total). The animals were randomly divided into two groups, control and treated, and raised from the hatchery stage to the end of the production cycle. The hens were kept in an intensive housing system in the same place and under the same controlled environmental conditions (24–32 °C, depending on the phase, and 50–70% humidity). Feed and water were supplied ad libitum.

### 4.3. Experimental Diets

Both groups were fed the same commercial formulation, according to the production phase, as routinely used at the farm. Feed of the treated group was supplemented with fermented defatted *alperujo* (FDA), which had first undergone a controlled anaerobic bacterial fermentation and was then defatted with chemical solvents (fat hydrolysis). Finally, it was desiccated at 80 °C in a low oxygen content atmosphere and then suffered a grinding process. FDA composition was determined by Labocor S.L. (Colmenar Viejo, Spain) (Table 3).

### 4.4. Production Performance

Productive parameters were registered for all production phases using the business management enterprise resource planning software Navision from Microsoft. The conversion rate was expressed as kg of feed consumed in relation to the number of eggs produced. Egg production and mortality were recorded daily; feed consumption was measured weekly. Eggs were collected over a 24 h period and weighed for determination of egg weight and grade.

### 4.5. Samplings and Necropsy

A total of 11 samplings across the three phases of the production cycle were performed during the study: hatchery (1–16 weeks old, four samplings), phase I (laying hens until peak laying, 16–23 weeks old, two samplings) and phase II (from 24-weeks old to the end of production, five samplings).

For all samplings, 15 hens of each group were sacrificed by the gas-stunning method. A complete necropsy was carried out for each animal. During the necropsy, a gross survey was performed, and duodenal and cecal samples were collected and fixed in a 4% formaldehyde-buffered solution for 48 h (Panreac Química SLU, Barcelona, Spain). In addition, fresh cecal feces were collected from each animal and preserved at −80 °C for the metagenomics analysis.

### 4.6. Histology

After fixation, intestinal tissues were dehydrated through an ethanol series and xylene substitute (Citadel 2000 Tissue Processor, Thermo Fisher Scientific, Waltham, MA, USA), then embedded in synthetic paraffin. After paraffin block formation (Histo Star Embedding Workstation, Thermo Fisher Scientific), 4-µm sections were cut (Finesse ME+ Microtome, Thermo Fisher Scientific), stained with hematoxylin and eosin (Panreac Química SLU) and mounted and examined under a light microscope (Leica, Wetzlar, Germany).

A histomorphometric study was performed using an image analyzer (Leica Application Suite, Leica), which measured different parameters of the duodenum and the cecum in each animal in five fields at 40× magnification. In each case, the number of duodenal villi and duodenal and cecal crypts were counted. Duodenal villi height was measured from the top of the villus to the villus–crypt junction; duodenal and cecal crypt depth was measured from the villus–crypt junction to the muscularis mucosae. A minimum of seven well-oriented villi and 14 crypts were measured from different sections of each hen.

### 4.7. Metagenomics

DNA was isolated from cecal samples using the QIAamp DNA Stool Mini Kit (Qiagen NV, Hilden, Germany). The next-generation sequencing and bioinformatics analyses of the bacterial 16S rRNA gene were performed by Stab Vida (Caparica, Portugal) and Era7 Bioinformatics (Granada, Spain). Regions V3 and V4 of 16S rRNA were sequenced on the Illumina Miseq platform using 300 bp paired-end sequencing.

### 4.8. Statistical Analysis

IBM SPSS Statistics Software (IBM; Armonk, NY, USA) was used for statistical analysis. Differences in production performance and histomorphological parameters between control and treated groups were assessed using Mann-Whitney test and statistical significance was considered at *p* < 0.05.

## 5. Conclusions

Dietary fermented defatted *alperujo* (FDA) supplementation in laying hens significantly improved duodenal villi height in pullets, which may enhance intestinal function during their productive life, as suggested by the significant decrease of broken eggs eliminated from production. Additionally, the increase of cecal crypt depth in pullets and aged hens appears to improves nutrient absorption. It may also influence non-specific immune responses by being able to better respond to potential harmful events, contributing to the intestinal health of laying hens. Our findings also suggest that diet composition can modulate intestinal microbiota at early life stages. Specifically, FDA supplementation seems to increase intestinal bacterial diversity by increasing the relative abundances of Firmicutes and Proteobacteria. Establishing a diverse microbiota early in life may enhance intestinal health by providing metabolic substances, improving immune response and competing with pathogenic bacteria, thus potentially reducing antimicrobial usage.

## Figures and Tables

**Figure 1 antibiotics-08-00215-f001:**
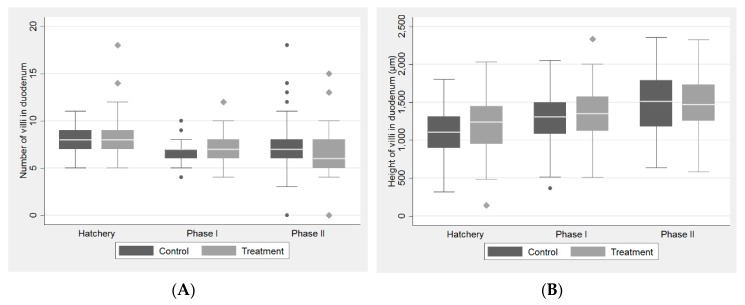
Graphs showing changes in histomorphometric parameters of the duodenal and the cecal mucosa from hatchery to phases I and II: (**A**) number of villi in the duodenum was significantly increased in phase I in the FDA-supplemented group (treatment); (**B**) height of villi in the duodenum was significantly increased in the hatchery in the FDA-supplemented group (treatment); (**C**) depth of crypts in the duodenum was significantly increased in the hatchery in the FDA-supplemented group (treatment); (**D**) number of crypts in the cecum: no significant differences were observed between the control and treatment group; (**E**) depth of crypts in the cecum was significantly increased in the hatchery and phase I in the FDA-supplemented group (treatment).

**Figure 2 antibiotics-08-00215-f002:**
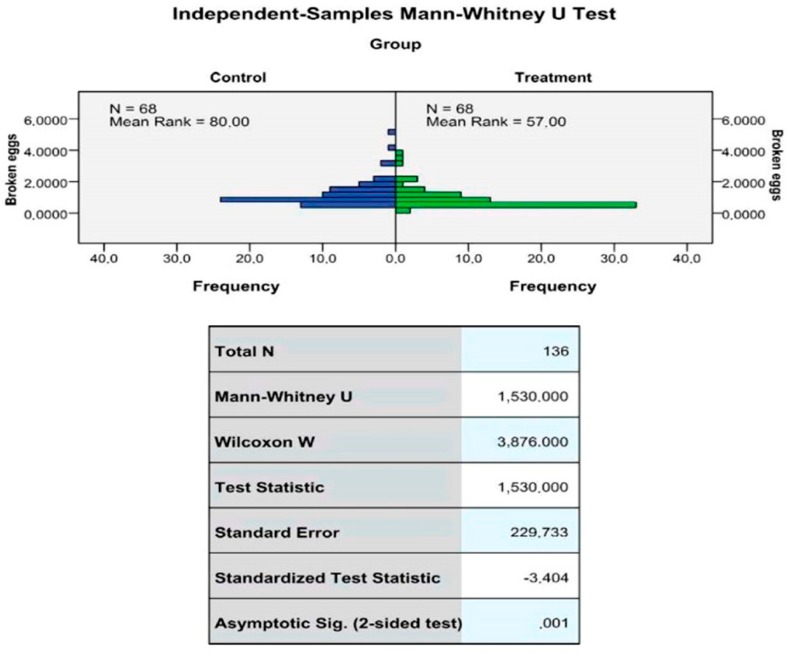
According to the Mann-Whitney U test, there is a statistically significant difference based on the *p*-value (Asymptotic Significance (two-sided test) = 0.001). The FDA-supplemented group (Mean Rank of 57.00) reported a lower number of broken eggs compared to control group (Mean Rank of 80.00).

**Figure 3 antibiotics-08-00215-f003:**
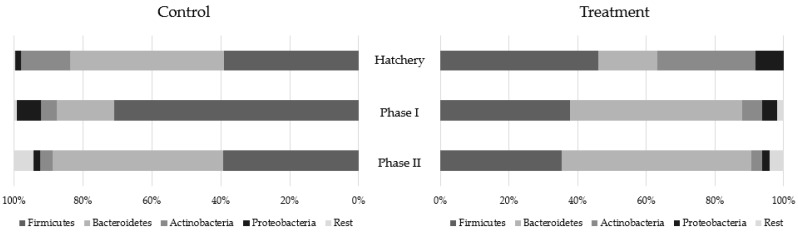
Metagenomic results. Phyla distribution by groups over time. Hens showed a higher relative abundance (RA) of Actinobacteria, Firmicutes and Proteobacteria in FDA-supplemented group (treatment) compared to control in hatchery. In phase I, the RA of Actinobacteria and Bacteroidetes was higher in the FDA group. In phase II, the RA of Bacteroidetes and Proteobacteria was higher in the treated group.

**Table 1 antibiotics-08-00215-t001:** Statistical results as mean and interquartile range (IQR) for villi number and height and crypt depth in the duodenum and crypt number and depth in the cecum in control and fermented defatted *alperujo* (FDA)-supplemented hens at three phases of production.

Histomorphometric Parameters	Control Group		FDA-Supplemented Group		*p*-Value ^1^
	Mean	[IQR]	Mean	[IQR]	
**Hatchery**					
Duodenum					
Number of villi	8.00	[2.00]	8.00	[2.00]	0.311
Height of villi (µm)	1109.95	[410.00]	1237.85	[492.10]	<0.001
Depth of crypts (µm)	161.05	[62.53]	188.10	[90.55]	<0.001
Cecum					
Number of crypts	14.00	[5.00]	15.00	[9.00]	0.028
Depth of crypts (µm)	325.65	[374.20]	358.15	[356.60]	<0.001
Production phase I					
**Phase I**					
Duodenum					
Number of villi	7.00	[1.00]	7.00	[2.00]	<0.001
Height of villi (µm)	1309.27	[411.03]	1353.32	[449.94]	0.157
Depth of crypts (µm)	249.66	[81.83]	255.15	[107.59]	0.135
Cecum					
Number of crypts	16.50	[16.00]	18.00	[17.00]	0.821
Depth of crypts (µm)	359.12	[411.14]	280.74	[440.91]	0.150
Production phase II					
**Phase II**					
Duodenum					
Number of villi	7.00	[2.00]	6.00	[3.00]	0.230
Height of villi (µm)	1511.99	[605.57]	1471.92	[476.69]	0.537
Depth of crypts (µm)	272.65	[138.47]	268.46	[102.79]	0.160
Cecum					
Number of crypts	14.00	[16.00]	14.00	[12.00]	0.883
Depth of crypts (µm)	402.68	[606.61]	518.50	[576.58]	<0.001

^1^ The Mann-Whitney test was used to assess significant differences (*p* < 0.05) between supplemented (*n* = 43) and control animals (*n* = 47).

**Table 2 antibiotics-08-00215-t002:** Production performance of control and FDA-supplemented hens.

Productive Parameter	Control Group	Treatment Group	*p*-Value ^1^
Mortality (%)	0.15	0.16	0.299
Laying (%)	78.00	79.00	0.970
Feed/hens (g)	115.37	113.37	0.124
Egg weight (g)	62.58	62.59	0.720
Egg mass (g/d)	48.04	49.28	0.730
Extra-large eggs (%)	5.20	5.40	0.989
Large eggs (%)	44.25	43.94	0.467
Medium eggs (%)	39.04	40.39	0.917
Small eggs (%)	5.8	5.4	0.084
Dirty eggs (%)	2.3	2.1	0.424
Broken eggs (%)	3.47	2.83	0.001
Total eggs (number)	21,526,722	21,892,058	0.808
Total eggs (Kg)	1,342,571.52	1,363,916.15	0.931
CI (Conversion Index)	0.37	0.35	-

^1^ The Mann-Whitney test was used to assess significant differences (*p* < 0.05) between supplemented (*n* = 43) and control animals (*n* = 47).

**Table 3 antibiotics-08-00215-t003:** Fermented defatted *alperujo* (FDA) composition.

Determination	Results
Moisture 103° (%w.w.)	12.2
Crude protein (Kjeldahl) (%w.w.)	6.4
Brute fat (%w.w.)	3.0
Ash content (%w.w.)	7.7
Lignin (%w.w.)	23.3
Acid detergent fiber (%w.w.)	39.2
Neutral detergent fiber (%w.w.)	49.3
Tannins (%w.w.)	0.06
Oleic acidity index (%w.w.)	46.1
Peroxide value (%w.w.)	7.9
Total polyphenols (meq/kg)	0.89
Crude fiber (%w.w.)	27.7

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
