# Peer review of "Effects on Intestinal Mucosal Morphology, Productive Parameters and Microbiota Composition after Supplementation with Fermented Defatted Alperujo (FDA) in Laying Hens"

_antibiotics, 2019, doi:10.3390/antibiotics8040215_

Round 1

Reviewer 1 Report

General Comments: The manuscript submitted by Agustín Rebollada-Merino et al. evaluated the morphology of the duodenal and cecal mucosa, the composition of the intestinal microbiota and the productivity of a batch of laying hens after FDO supplementation. The authors demonstrate that microbiota composition in the hatchery group supplemented with FDO had a higher abundance of Actinobacteria, Firmicutes, and Proteobacteria, and higher bacterial diversity, Moreover, it was shown that there were significant differences in the number of eggs marketed from the supplemented group during the production period.

There are some items in the manuscript that should be clarified. The following comments and questions are included in order to help the authors to improve the quality of the paper.

Minor Comments:

Lines 97, 112, and 128: A figure should be able to stand alone. Readers want to understand it without going back and forth between the figure and the text sections. Please make your figure legends more clear.

Minor Comments:

1. Lines 266-277: Put part of “5. Conclusions” ahead of Line 203 “4. Materials and Methods”.

2. Lines 91-92: Put the title of Table 1 together with the table.

3. Line 96: It’s better to make sure all the pictures of Figure 1 on one page.

4. Line 109: The last line needs a horizontal line if Table 2 is divided.

Author Response

We thank reviewer 1 for the very thorough critique and we have implemented the comments made.

Lines 97, 112, and 128: A figure should be able to stand alone. Readers want to understand it without going back and forth between the figure and the text sections. Please make your figure legends more clear. This information has now been modified accordingly.

Lines 266-277: Put part of “5. Conclusions” ahead of Line 203 “4. Materials and Methods”. This section has been moved.

Lines 91-92: Put the title of Table 1 together with the table. The title has been moved together with the table.

Line 96: It’s better to make sure all the pictures of Figure 1 on one page. This question has been addressed.

Line 109: The last line needs a horizontal line if Table 2 is divided. This has been changed.

Reviewer 2 Report

Lines 36-37. Please provide year. The article will be read for many years and the term “last season” is not appropriate. Line 57. Supplier The authors use fermented defatted olives FDO, alperujo and olive oil by product for the same term. Please use only one in the whole manuscript in order to avoid misunderstandings. Fermented defatted olives in my opinion is not appropriate for the olive oil industry solid waste. There is no discussion in the introduction regarding the waste used in the present study. In addition the authors present works that used olive oil for feed supplement that is irrelevant with the present study. Please revise the introduction and present only relevant works. Lines 218-219. Only this? Provide more details about the preparation of the alperujo. Any reference? Details of fermentation etc. Provide an analysis of the alperujo given to the hens. Line 102. dietary FDO?? New term? There is no discussion regarding the productive parameters. Why? Discussion is very general and the authors do not try to provide an explanation about the results. Why were these results observed? This is very important in a research article. The authors compare their results with those of other studies. However there is no explanation why these results were observed. An analysis of FDO is necessary and the authors should try to correlate their results with this composition. Lines 105-107. The authors, although the results were not significantly different, concluded that the new feed resulted in 1.7% increase in egg sale profits and a 1.5% decrease in feed costs. What about the increase in mortality? This is also no significant but it had not any effect? Line 269. “significant increase in the number of eggs marketed”. Where is this result? In general the introduction should be rewritten and focus on FDO. The discussion should be rewritten and provide possible explanations. An analysis of fermented FDO and details of this fermentation should be provided.

Author Response

We thank reviewer 2 for the comments, trying to address all the questions proposed.

Lines 36-37. Please provide year. The article will be read for many years and the term “last season” is not appropriate.

The productive cycle of olive oil production was expressed in campaigns (from October-November to September-October). This information was added to the text (line 37-39).

Line 57. The authors use fermented defatted olives FDO, alperujo and olive oil by product for the same term. Please use only one in the whole manuscript in order to avoid misunderstandings. Fermented defatted olives in my opinion is not appropriate for the olive oil industry solid waste.

We have considered the correct term would be fermented defatted alperujo (FDA), as the raw material, alperujo, has undergone fermentation and fat hydrolysis (defatted). The term olive was, in fact, not appropriate. The text has changed accordingly.

There is no discussion in the introduction regarding the waste used in the present study.

We have explained that alperujo “per se” is not adapted to animal feed and therefore modifications of the raw material are needed to avoid counterproductive effects: “Alperujo, like other olive oil by-products, is a fat-rich compound. However, although possess beneficial effects, fat-rich compounds should be limited in the feed formulation to avoid their counterproductive effects in high percentages [14]. For this reason, in this study alperujo had first undergone a fermentation process to stabilize the raw material, a hydrolysis to decrease the total fat content and grinding process and desiccation in order to adapt it to animal feed, obtaining fermented defatted alperujo (FDA).” This information was included in the text (lines 66-71).

In addition the authors’ present works that used olive oil for feed supplement that is irrelevant with the present study. Please revise the introduction and present only relevant works.

We thought these works were relevant to justify the limited use of olive oil by-product on hens feed, although we have considered reviewer comments and that information was removed from the text.

Lines 218-219. Only this? Provide more details about the preparation of the alperujo. Any reference? Details of fermentation etc. Provide an analysis of the alperujo given to the hens.

Details about FDA preparation and composition are provided. There are not available references due to the innovative procedure (this product is being evaluated to be officially registered). The information was included in Material and methods section (lines 257-264).

Line 102. Dietary FDO?? New term?

It has been modified taking into account comment number 2.

There is no discussion regarding the productive parameters. Why? Discussion is very general and the authors do not try to provide an explanation about the results. Why were these results observed? This is very important in a research article. The authors compare their results with those of other studies. However there is no explanation why these results were observed. An analysis of FDO is necessary and the authors should try to correlate their results with this composition.

We have considered your advices and improved the discussion on what regards production.

Lines 105-107. The authors, although the results were not significantly different, concluded that the new feed resulted in 1.7% increase in egg sale profits and a 1.5% decrease in feed costs. What about the increase in mortality? This is also no significant but it had not any effect?

We have included that the mortality rate was higher but not significant in the treated group. However, we have discussed only significant results (broken eggs), as is the only parameter with statistical significance between groups.

Line 269. “significant increase in the number of eggs marketed”. Where is this result?

We have considered the term eggs produced instead of eggs marketed, which is less appropriate.

In general the introduction should be rewritten and focus on FDO. The discussion should be rewritten and provide possible explanations. An analysis of fermented FDO and details of this fermentation should be provided. This information was included in the new draft of the manuscript.

Round 2

Reviewer 1 Report

The authors have revised the manuscript and modified relevant elements in order to answer the previous comments, with considerable improvement. The findings of this article are important to those with closely related research interests and should be published.

Author Response

We thank reviewer 1 for the comments/suggestions made. We do think that the article has been improved after the revision made.

Reviewer 2 Report

Lines 303-305. Provide more details in statistical analysis. For example which tests were used?  Table 2. Add statistical analysis on table (for example p value, ± values etc) Avoid the extensive use of references for a single sentence. For example, lines 44-46: "As a result of this process, a solid by-product called two-phase olive mill waste, sometimes referred to as olive cake or olive pulp but commonly known as alperujo, is obtained [5–7]". Please explain the use of 3 references. Revise and use only one. Probably 7 which is also used in the next sentence.  What about this study? Animals 20199(7), 427 Whole Blood Transcriptome Analysis Reveals Positive Effects of Dried Olive Pomace-Supplemented Diet on Inflammation and Cholesterol in Laying Hens. This article should be also added and discussed. Table 3. Delete % after results Line 258. "controlled anaerobic bacterial fermentation". Add details about the fermentation (microorganisms, temperature, time etc). It was also commented in my previous review, however, no answer by the authors.

Author Response

Lines 303-305. Provide more details in statistical analysis. For example which tests were used?  We appreciate your comments and suggestions. We have added details required in statistical analysis performed using the Mann-Whitney test.

Table 2. Add statistical analysis on table (for example p value, ± values etc) We appreciate your comments and suggestions. P-values has been added on table 2.

Avoid the extensive use of references for a single sentence. For example, lines 44-46: "As a result of this process, a solid by-product called two-phase olive mill waste, sometimes referred to as olive cake or olive pulp but commonly known as alperujo, is obtained [5–7]". Please explain the use of 3 references. Revise and use only one. Probably 7 which is also used in the next sentence.  We appreciate your comments and suggestions.

What about this study? Animals 20199(7), 427 Whole Blood Transcriptome Analysis Reveals Positive Effects of Dried Olive Pomace-Supplemented Diet on Inflammation and Cholesterol in Laying Hens. This article should be also added and discussed. We have considered this paper. It contains valuable information that supports the employment of olive by-product DOP in animal feed. Despite this, it is focused on gene expression and cholesterol content modifications after DOP supplementation. It has valuable information to support and complete our study, but we cannot compare it with ours because of the different methodology employed as we have not evaluated egg composition and neither a gene expression analysis has been performed. Thank you for your advices, the paper suggested would be taken into account in current and future researches following olive oil by-product revalorization by our research unit.

Table 3. Delete % after results. We appreciate your comments and suggestions

Line 258. "controlled anaerobic bacterial fermentation". Add details about the fermentation (microorganisms, temperature, time etc). It was also commented in my previous review, however, no answer by the authors. We apologize for not being so clear with this in the first revision. In the first round we have added details about FDA preparation and the chemical composition of the product. Unfortunately, the process will be registered by the company that provide the FDA (is under revision now) and, at this point, there are not available references of the fermentation procedure.